# Surface Segregation Process and Its Influence on High-Temperature Corrosion of Iron-Based Alloys Containing Aluminium, Vanadium, Titanium and Germanium

**DOI:** 10.3390/ma18030557

**Published:** 2025-01-26

**Authors:** Magdalena Sobota, Karolina Idczak, Robert Konieczny, Rafał Idczak

**Affiliations:** Institute of Experimental Physics, University of Wrocław, pl. M. Borna 9, 50-204 Wrocław, Poland; magdalena.sobota@uwr.edu.pl (M.S.); karolina.idczak@uwr.edu.pl (K.I.); robert.konieczny@uwr.edu.pl (R.K.)

**Keywords:** corrosion, Mössbauer spectroscopy, X-ray photoelectron spectroscopy, iron alloys

## Abstract

The surface segregation process and its influence on high-temperature corrosion of five alloys—Fe_0.95_Al_0.05_, Fe_0.95_V_0.05_, Fe_0.90_Al_0.05_V_0.05_, Fe_0.95_Ti_0.05_ and Fe_0.95_Ge_0.05_—were studied using X-ray photoelectron spectroscopy (XPS) and ^57^Fe Transmission Mössbauer Spectroscopy (TMS). To prepare the alloys with the highest surface concentration of solutes, the samples were annealed at elevated temperatures to induce the surface segregation process. After that, they were exposed to air at 870 K for 1 and 5 h. It was found that the Fe_0.95_Ti_0.05_ sample annealed at 1073 K had much better anti–corrosion properties than other alloys studied. This finding can be correlated with the extremely high concentration of titanium on the surface, which was more than four times that of iron. In contrast to other alloys studied in this work, the passive layer formed on the surface of Fe_0.95_Ti_0.05_ greatly enhanced its resistance to corrosion.

## 1. Introduction

Corrosion of iron and its alloys is a fundamental industrial and scientific problem that costs the economies of most countries enormous amounts of money every year. Finding a solution to this problem is still a major challenge in materials science and engineering. There are several strategies for corrosion prevention and protection methods, including protective coatings, corrosion-resistant alloys, corrosion inhibitors and cathodic and anodic protection [1]. By applying these methods, the negative effects of corrosion can be reduced, thus ensuring the longevity and reliability of metal structures and components.

In our previous works, we used ^57^Fe Transmission Mössbauer Spectroscopy (TMS) to study the corrosion in iron [2] and iron alloys such as Fe–Cr [3], Fe–Si [4], Fe–Ni–Cr, Fe–Ni–Si [5] and Fe–Cr–Si [6] with various concentrations of solutes to determine the optimal chemical composition and the optimal method of synthesis for the preparation of corrosion-resistant alloys. It was found that among the materials studied, the most promising corrosion-resistant ones are iron-based ternary alloys containing chromium and silicon. Further investigation has shown that their excellent anti-corrosion properties are due to the surface segregation of Cr and Si atoms, which form the passive layer at the alloy surface [7,8]. These interesting findings inspire us to continue investigating the surface segregation phenomenon and its influence on high-temperature corrosion in iron-based alloys.

In this work, we chose iron-based alloys containing aluminium, vanadium, titanium and germanium. According to the Ellingham diagram, which describes the Gibbs free energy change ($\Delta G^{O}$) for each oxidation reaction as a function of temperature, $\Delta G^{O}$ values for Al, V and Ti in the temperature range of 300 K–800 K are far below those for Fe [9]. This means that the Al, V and Ti oxides are more stable than iron oxides such as FeO, Fe_2_O_3_ and Fe_3_O_4_. Moreover, metal associated with lower Gibbs free energy of oxidation on the diagram will reduce the oxide with the higher Gibbs free energy of formation. Taking this into account, it is possible to consider a scenario in which the surface of the studied alloys is covered with a significant quantity of Al, V or Ti atoms via the surface segregation process. Consequently, due to the selective oxidation of these elements, the passive layer, which acts as a barrier against corrosion damage, is formed. In the case of Ge, the value of $\Delta G^{O}$ is comparable with that for Fe [10].

Our choice was also influenced by the fact that the surface segregation of one component in binary metal alloys can be observed or predicted for most transition metal alloys that form disordered binary substitutional alloys [11,12]. The selected Fe–Al, Fe–V and Fe–Ge systems are characterized by a wide α-Fe solid solution range [13,14,15]. Therefore, the addition of Al, V or Ge to Fe forms a disordered substitutional alloy, which crystallizes in a single-phase body-centred cubic lattice (bcc). In the case of the Fe–Ti system, according to a phase diagram given by Okamoto [16], the solution of Ti in the α-Fe matrix is limited to about 3–5 at.%. However, our earlier study of iron-based Fe–Ti alloys showed that the Fe_0.95_Ti_0.05_ sample prepared by an arc-melting method was a homogenous bcc alloy [17].

Finally, it should be noted that iron-based alloys containing aluminium, vanadium, titanium and germanium are already widely used in modern industry. In particular, iron–aluminium alloys offer an attractive combination of excellent oxidation and sulfidation resistance with low cost and acceptable tensile strength [18]. In addition, the aluminium–iron oxide thermite reaction is a simple reduction of iron oxide using Al to metallic Fe and aluminium oxide [19]. Vanadium increases strength, toughness and weldability in high-strength low-alloy steels (HSLAs) [20]. Here, it should also be noted that vanadium is a strong carbide-forming element [21]. In the case of titanium, it has superior corrosion resistance in many aggressive chemical environments and has the best specific mechanical properties among engineering metals (i.e., mechanical properties divided by density) [22,23]. Therefore, its use in the production of anti-corrosion alloys seems reasonable. Lastly, germanium is widely used in many fields like communications, electronics, aerospace, solar cells and metallurgy [24].

Our research was carried out in two stages. In the first one, for each studied alloy, the optimal temperature (OT) for the most efficient surface segregation of the solutes was determined by using X-ray photoelectron spectroscopy (XPS) measurements. In the second stage, to demonstrate the correlation between surface segregation and corrosion resistance, the samples after heating at the optimal temperature as well as at the highest temperature of 1273 K were subjected the oxidation process and their anti-corrosion properties were examined by TMS. The results obtained are discussed in detail and compared with the data from earlier works.

## 2. Materials and Methods

### 2.1. Sample Preparation and Oxidation

The Fe_0.95_Al_0.05_, Fe_0.95_V_0.05_, Fe_0.95_Al_0.05_V_0.05_, Fe_0.95_Ti_0.05_ and Fe_0.95_Ge_0.05_ samples were prepared using the arc-melting technique in Ti-gettered Ar atmosphere and then cold-rolled to a thickness of about 70 μm. The detailed procedure is described in our previous papers [8,25]. The initial chemical composition of the samples was confirmed by weighing the resultant ingots. The weight loss after melting did not exceed 0.3%. In the next step, the first group of samples, marked as ‘–XPS’, was placed in an ultra-high vacuum (UHV) chamber and subsequently annealed at elevated temperatures from 823 K to 1273 K for 15 min to induce the segregation process. After determining the optimal temperature for the most efficient surface segregation of the solutes, the second group of samples, marked as ‘–OT’, was heated in a vacuum furnace (dynamic pressure was lower than 10^−5^ mbar) at the selected optimal temperature, and then both series were subjected to an oxidation process at 870 K in air, for selected periods of time (in cycles). After each step of oxidation, the TMS measurements were performed.

### 2.2. Measurements and Data Analysis

The XPS technique was used to investigate the chemical composition and bonding environment of the surface region of the alloys studied. For these types of samples, the typical sampling depth for which the XPS signal can be detected is equal to 3.95 nm. XPS measurements were performed in the UHV chamber, equipped with an SPECS Phoibos 150 hemispherical analyser with standard Mg and Al Kα X-ray sources. All scans were taken at a photoelectron take-off angle of 90° at RT under a pressure lower than $10^{-8}$ Pa, after a standard calibration procedure (for Au 4f doublet values for the clean Au sample). The measured spectra were analysed using the CasaXPS software (http://www.casaxps.com/), The background of the spectra was subtracted using a Shirley method and the deconvolution of the spectra was made by the Gaussian–Lorentzian (GL (30%)) fitting method, followed in accordance with the expected atomic bonds on the surface, limited to the full width at half maximum (FWHM <2.8 eV) of the peaks. In the case of Fe 2p spectra analysis, the Gupta–Sen multiplet peaks (GS) fitting method was used [26]. The surface atomic concentration of each element $c_{i}$ was calculated using the following formula:(1)$$\begin{array}{c} c_{i}=\frac{\frac{I_{i}\lambda_{i}}{\sigma_{i}}}{\sum_{i}\frac{I_{i}\lambda_{i}}{\sigma_{i}}}\cdot 100\%, \end{array}$$
where *I* denotes a selected XPS peak intensity, λ is a Scofield parameter [27] and λ is an inelastic mean free path of an electron with a certain kinetic energy related to the XPS core-level line [28]. The presented $c_{i}$ values have an uncertainty less than 2%.

In order to investigate the alloys’ corrosion resistance, the selected samples were exposed to air at 870 K for different time intervals. After each step of oxidation treatment, the samples were studied by TMS. The spectra were measured in transmission geometry with a conventional constant-acceleration spectrometer, using a 3.7 GBq ^57^Co–in–Rh standard source with a full width at half maximum (FWHM) of 0.22 mm/s. Each measured TMS spectrum was analysed using a least-squares fitting procedure in terms of a sum of different numbers of six-line patterns (sextets) corresponding to various isomer shifts (IS), quadrupole shifts (QS) and hyperfine fields (*B*) at ^57^Fe nuclei related to different chemical states of ^57^Fe Mössbauer probes. The fitting procedure was performed under the thin absorber approximation. For each sextet, the two-line area ratio $I_{16}/I_{34}$ was constant and equal to 3/1. The ratio $I_{25}/I_{34}$ as well as the three linewidths $\Gamma_{16}$, $\Gamma_{25}$ and $\Gamma_{34}$ were free parameters. All the IS values presented in this paper are related to the IS value determined for α–Fe at room temperature.

The TMS spectra measured for the substitutional iron alloys that crystallise in a body-centred cubic structure (bcc) can be well described by the additive model [29]. The model assumes that the influence of solute atoms on the hyperfine parameters IS and *B* of a subspectrum is additive and independent of the atom positions in the given coordination shell of the ^57^Fe, although it can be different for atoms located in unlike shells. Therefore, for each subspectrum, the quantities *B* and IS can be described as linear functions of the numbers $n_{1}$ and $n_{2}$ of solute atoms located, respectively, in the first and second coordination shells of the Mössbauer probe. These functions can be described as follows:(2)$$\begin{array}{c} IS\left(n_{1},n_{2}\right)=IS_{0}+n_{1}\Delta IS_{1}+n_{2}\Delta IS_{2},\\ B\left(n_{1},n_{2}\right)=B_{0}+n_{1}\Delta B_{1}+n_{2}\Delta B_{2}, \end{array}$$
where $\Delta IS_{1}\left(\Delta B_{1}\right)$ and $\Delta IS_{2}\left(\Delta B_{2}\right)$ denote changes in IS(B) with one solute atom in the first and second coordination shell of the Mössbauer atom, respectively. Furthermore, it was assumed that the QS in the cubic lattice is equal to zero [30].

The hyperfine parameters derived from the additive model and used to describe the measured TMS spectra of the bcc iron alloys are listed in Table 1. In the case of the Fe–Al [31] and Fe–Ge [32] alloys, it was assumed that Δ*B*_2_ = 0 T and ΔIS_2_ = 0 mm/s. For the Fe–V, Fe–Ti and Fe–Al–V alloys, it was assumed that Δ*B*_1_ = Δ*B*_2_ and ΔIS_1_ = Δ*IS*_2_ [17,29].

Finally, the relative intensities *I* of the components for each Möossbauer spectrum were determined. Since the relative ratio of the Lamb–Mössbauer factors $f_{\alpha -\mathrm{Fe}}$:$f_{{\mathrm{Fe}}_{2}O_{3}}$:$f_{{\mathrm{Fe}}_{3}O_{4}}$ is close to 1:1.08:1.05 [33], the fraction $c_{i}$ of absorbing atoms corresponding to the component *i* can be computed using $I_{i}$ and Equation (Equation 3):(3)$$\begin{array}{c} c_{i}=\frac{I_{i}/f_{i}}{\sum_{i}I_{i}/f_{i}}\cdot 100\%. \end{array}$$

The determined *c* values were used to find the parameters c(alloy), $c({\mathrm{Fe}}_{2}O_{3})$ and $c({\mathrm{Fe}}_{3}O_{4})$, which are the fractions of absorbing ^57^Fe atoms located in the bcc iron alloys, as well as in α-${\mathrm{Fe}}_{2}O_{3}$ and ${\mathrm{Fe}}_{3}O_{4}$ oxides, respectively.

## 3. Results and Discussion

### 3.1. Fe–Al Alloy

The first step was to determine the optimal temperature for the most efficient aluminium segregation process using XPS measurements. It should be noted here that XPS spectra reveal the presence of Fe, Al, C and O at the surface. The presence of C and O atoms can be connected with the fact that before XPS measurement in UHV conditions, the sample had direct contact with atmospheric gases, which led to surface contamination. Table 2 shows ratios of the surface atomic concentration of aluminium to iron c*_Al_*/c*_Fe_* and oxygen to alloy constituents c*_O_*/c_*(Fe+Al)*_ obtained from XPS data for the Fe_0.95_Al_0.05_ sample after annealing at temperatures from 823 K to 1273 K for 15 min in UHV. The first measurable c*_Al_*/c*_Fe_* ratio is obtained for the sample annealed at 873 K and already at this temperature, this ratio is much higher than the value of 0.05 calculated for the sample’s bulk. The highest c*_Al_*/c*_Fe_* ratio is achieved after annealing at 933 K, which is 1.13—more than 22 times higher than 0.05. Therefore, it can be stated here that a strong surface segregation of Al occurred in the Fe_0.95_Al_0.05_ alloy. Further sample heating leads to decrease in the c*_Al_*/c*_Fe_* ratio; however, even after annealing at 1273 K, this ratio is still much higher than the bulk one. In the case of the c*_O_*/c_*(Fe+Al)*_ ratio, a parameter that corresponds to the presence of oxygen in the surface region, it can be seen that the first significant decrease of this ratio is observed after annealing at 873 K, and the second at 1073 K. This result may suggest that the process of oxygen-induced surface segregation occurs in a way that is similar to previously investigated Fe–Cr–Si alloys [8]. Taking into account the obtained XPS data, it can be stated that the optimal temperature for the surface segregation process of Al solutes in Fe_0.95_Al_0.05_ is close to 933 K.

A detailed analysis of the XPS spectra measured for the Fe_0.95_Al_0.05_ sample after annealing at 933 K/15 min and 1273 K/15 min in UHV is presented in Figure 1. In the Al 2p core-level line (see Figure 1a), two peaks can be distinguished—at the binding energy (BE) values of 73.0 eV and 75.4 eV—and they can be assigned to Al^0^ and Al^3+^ species, respectively [34]. In particular, the peak with a BE of 75.4 eV suggests the presence of Al_2_O_3_ as well as the mixed oxide FeAl_2_O_4_ [34]. In the Fe 2p_3/2_ spectrum, three components are observed (see Figure 1d): metallic Fe^0^ at a BE of 707.1 eV, and two oxide peaks at 708.6 eV (Fe^2+^) and at 710.5 eV (multiplet of Fe^2+^ and Fe^3+^) [2,35]. The spectrum of XPS C 1s is composed of two peaks at the BEs of 284.9 eV (C–C bond) and 286.5 eV (C–O bond) [36]. Here, it should be noted that no interaction between carbon and alloy constituents is observed. In the O 1s spectrum, after analysis, two components were distinguished at the BEs of 532.1 eV and 533.6 eV (see Figure 1c), which represent various oxygen species such as metal oxides (O–Me) and carbon oxides present at the surface and defined in other XPS spectra [35,37,38].

After annealing at a temperature of 1273 K, the surface region of the Fe_0.95_Al_0.05_ sample was mainly composed of iron and aluminium. In the XPS Fe 2p_3/2_ high-resolution region, the same components as after annealing at 933 K are distinguished (see Figure 1d); however, the metallic Fe component shows the highest intensity and its contribution increases the most. The atomic concentration of Al slightly increases and the positions of the components shift to the higher BE range at 73.0 eV and 75.4 eV; however, the peak of aluminium oxide still dominates (see Figure 1a). The signals from oxygen and carbon decrease significantly. In the case of C 1s, the peak position is slightly shifted to a lower BE, while for the O 1s envelope, the peak position shifts to a higher BE.

In the next step, two samples of Fe_0.95_Al_0.05_ were subjected to oxidation in air at 870 K. The first one was a sample that was used for XPS measurements, marked as –XPS. The second sample, marked as –OT, was additionally prepared by arc-melting and heated in vacuum at 933 K/2 h.

The TMS spectra of both samples were recorded at room temperature and are presented in Figure 2. In the case of the as–prepared sample (AP), four sextets can be observed. According to the additive model mentioned above, all these components can be assigned to the ^57^Fe nuclei in the bcc Fe_0.95_Al_0.05_. After XPS measurements or heating at the optimal temperature, no significant changes in the collected spectra were observed. In the TMS spectra of the –XPS sample oxidised at 870 K/1 h, three additional sextets can be seen. One with hyperfine parameters *B* = 51.7(1) T, IS = 0.364(9) mm/s and QS = −0.114(9) mm/s can be assigned to ^57^Fe nuclei in hematite Fe_2_O_3_ and the other two with $B_{1}$ = 49.1(2) T, $IS_{1}$ = 0.269(17) mm/s and $QS_{1}$ = 0.071(17) mm/s and $B_{2}$ = 46.1(9) T, $IS_{2}$ = 0.670(13) mm/s and $QS_{2}$ = −0.028(13) mm/s to magnetite Fe_3_O_4_ [2]. After 5 h of oxidation, the number of sextets does not change. For the –OT oxidised for 1 h and 5 h, an additional sextet assigned to the Fe_2_O_3_ was observed. The content of iron oxides in each sample is listed in Table 3. For –XPS after 1 h of oxidation, the c(Fe_2_O_3_) = 6.5% and c(Fe_3_O_4_) = 6.1%. After an additional 4 h of heating in air, the concentrations increased to 7.4% and 8.8%, respectively. For the –OT sample, the c(Fe_2_O_3_) increased from 3.4% to 5.2%. This result clearly shows that the sample prepared by heating at an optimal temperature of 933 K has much better anti-corrosion properties than the alloy after thermal treatment at 1273 K.

### 3.2. Fe–V Alloy

The concentration ratios of c*_V_*/c*_Fe_* and c*_O_*/c_*(Fe+V)*_ are listed in Table 4. The c*_V_*/c*_Fe_* increases during heating in the UHV to 1.25 at 973 K and decreases for higher annealing temperatures. Compared to the bulk, the concentration at this temperature is about 25 times higher. In the case of the c*_O_*/c_*(Fe+V)*_ ratio, it decreases drastically after the first (823 K) and second (873 K) annealing phases, while at higher temperatures, the decrease slows down significantly. Similar to Fe–Al, in this sample, the surface segregation is strongly enhanced by the presence of oxygen. Taking into account the obtained XPS data, it can be stated that the optimal temperature for the surface segregation process of vanadium solutes in Fe_0.95_V_0.05_ is close to 973 K.

Figure 3 shows XPS spectra measured for the Fe_0.95_V_0.05_ sample heated at the optimal 973 K and the highest 1273 K temperature for 15 min. Three peaks, in the C 1s spectrum (see Figure 3a), are distinguished: V–C at the BE of 282.9 eV [39], C–C at the BE of 284.8 eV and C–O at the BE of 286.4 eV BE. In the case of vanadium, two doublets were observed (see Figure 3b) with the maxima at the positions of 513.4 eV and 521.0 eV and 515.2 eV and 523.0 eV, respectively. The first doublet corresponds to the metallic V^0^ and the second to V^5+^ species [40]. The presence of oxides was also detected in the Fe 2p signal (see Figure 3d), where three components can be defined: one metallic Fe at the BE of 707.0 eV and two oxidised Fe placed at the positions of 708.6 eV and 710.0 eV. One additional component for the 1273 K sample can be seen at 711.7 eV. The O 1s core-level line has a broad and non-symmetric shape (see Figure 3c), where three components can be distinguished: 531.2 eV assigned to iron and vanadium oxides (O–Me) [41,42], double-bond O=C at 532.7 eV and other non–stoichiometric oxides (O_*x*_) at the BE of 533.8 eV. The first component splits into two after additional annealing at 1273 K: one at the BE of 530.4 eV for O–Fe, and the second at the BE of 531.4 eV corresponding to O–V.

Similar to the Fe_0.95_Al_0.05_ sample, the Fe_0.95_V_0.05_ –XPS and –OT were oxidised under atmospheric conditions at 870 K. The collected TMS spectra are presented in Figure 4, while the obtained hyperfine parameters are listed in Table 1. In the case of the as-prepared sample, four sextets assigned to iron atoms located in the bcc Fe–V alloy can be seen. After annealing in the UHV or heating at 973 K/2 h, no additional components were observed. After 1 h of oxidation in both samples –XPS and –OT, the component assigned to Fe_2_O_3_ could be observed and after 5 h, two sextets related to Fe_3_O_4_ were found. For the –XPS sample, the amount of Fe_2_O_3_ increased from 4.5% to 10.1% and after 5 h of oxidation, the concentration of Fe_3_O_4_ was equal to 3.5% (Table 3). For the –OT sample oxidised for 5 h, the concentrations of iron oxides were equal to 5.3% and 1.3%, respectively.

This result leads to similar conclusions as for the Fe_0.95_Al_0.05_ alloy, i.e., that heating the sample at an optimal temperature of 973 K results in improvement in its anti-corrosion properties. However, comparing the data presented in Table 3, one can notice that the amount of iron oxides in bulk of the Fe_0.95_V_0.05_ –OT sample oxidised for 1 h and 5 h was higher than for Fe_0.95_Al_0.05_ –OT. Therefore, the Fe_0.95_V_0.05_ alloy is less corrosion-resistant than Fe_0.95_Al_0.05_.

### 3.3. Fe–Al–V Alloy

Table 5 presents the surface concentration ratios for each solute separately. The highest c*_V_*/c*_Fe_* and c*_Al_*/c*_Fe_* ratios are achieved after annealing of the sample at 1273 K. This finding suggests that the segregation of solutes in this alloy (two solutes) differs from that for the previous ones (one solute). When comparing ratios for aluminium, the efficiency of segregation is much higher in the Fe–Al–V alloy than Fe–Al. The estimated ratios are much higher for all heating temperatures. In contrast, the vanadium ratios are higher for the Fe–V alloy than for the Fe–Al–V sample, with the only exception for 1273 K. Moreover, in this sample, the c*_O_*/c_*Fe+Al+V*_ ratio significantly decreases already after annealing at 873 K. This result may suggest that in this alloy, the surface segregation is induced mainly by temperature.

XPS spectra of the Fe_0.90_Al_0.05_V_0.05_ sample, annealed at 1273 K/15 min, are presented in Figure 5. In the V 2p spectrum, three components are distinguished. The first doublet, with the highest intensity at the BEs of 513.4 eV and 520.8 eV represents V^0^, while the other two at the BEs of 514.9 eV and 522.0 eV and 516.2 eV and 523.1 eV correspond to V^5+^. In the Al 2p core-level line, two peaks representing Al^3+^ species are detected (75.2 eV and 76.3 eV). The Fe 2p_3/2_ spectrum is deconvoluted into three components: the main Fe^0^ at the BE of 707.0 eV and some oxidation states represented by two components at the positions of 708.5 eV and 710.2 eV. In the case of C 1s, three peaks were distinguished: vanadium or/and aluminium carbide at 282.8 eV (Me–C) [39,43], a carbon–carbon single bond at 284.6 eV and a carbon–oxygen double bond at 286.2 eV. In the O 1s spectrum, one component was distinguished at 532.6 eV, which corresponds to non-stoichiometric metal oxides.

TMS spectra of the Fe_0.90_Al_0.05_V_0.05_ –XPS sample are presented in Figure 6. As was pointed out, in this sample, the surface segregation of the solutes is caused mainly by temperature, not by adsorbed oxygen. Therefore, in this case, the oxidation experiment was conducted only for the sample after annealing at the highest temperature of 1273 K. For the as-prepared and –XPS samples, four sextets are observed and they are related to Fe atoms located in the Fe–Al–V alloy bcc. After 5 h of oxidation, the additional sextet could be seen and was assigned to the hematite with c(Fe_2_O_3_) = 2.7% (Table 3). As can be seen, the fraction of iron atoms located in iron oxides observed for the Fe_0.90_Al_0.05_V_0.05_ sample after oxidation for 5 h is much lower than in the case of the Fe-V and Fe–Al alloys. However, in contrast to Fe_0.90_Al_0.05_ and Fe_0.90_V_0.05_, this alloy contains 10 at.% of solutes and it is more appropriate to compare its anti-corrosion properties with the Fe_0.90_Cr_0.10_ and Fe_0.90_Cr_0.05_Si_0.05_ alloys studied in [8]. According to data presented in our earlier work, one can conclude that the corrosion resistance of the Fe_0.90_Al_0.05_V_0.05_ alloy is comparable to that observed for Fe_0.90_Cr_0.10_ and much worse than for Fe_0.90_Cr_0.05_Si_0.05_.

### 3.4. Fe–Ti Alloy

The estimated surface concentration ratios c*_Ti_*/c*_Fe_* and c*_O_*/c*_Fe+Ti_* for the Fe_0.95_Ti_0.05_ alloy are presented in Table 6. Here, the highest c*_Ti_*/c*_Fe_* = 4.34 was achieved after annealing of the sample at 1073 K. This ratio is more than 86 times higher than the bulk value (0.05). Therefore, it indicates an extraordinary high surface segregation efficiency for titanium atoms.

Figure 7 presents XPS spectra of the Fe_0.95_Ti_0.05_ sample heated at 1073 K and 1273 K for 15 min in the UHV. After annealing at the optimal temperature for segregation (1073 K), the Ti 2p spectrum was composed of three doublets. The main, with the highest intensity, was the metallic Ti placed at the BEs of 455.2 eV and 460.9 eV. The other two, representing oxidised Ti^3+^ and Ti^4+^, were located at the BEs of 456.6 eV and 462.1 eV and 457.8 eV and 463.3 eV, respectively [44]. The Fe 2p_3/2_ was composed of three peaks: metallic iron at a BE of 707.1 eV and oxidised iron placed at 708.7 eV and 710.5 eV. Similar to previous alloys, carbon and oxygen signals were also analysed. In the C 1s spectrum, three components were distinguished: Ti–C at 282.1 eV [45], a C–C single bond at 284.6 eV and a C–O at 286.7 eV BE. In the O 1s spectrum, two components corresponding to O–metal (O–Me) and O–C compounds were observed. The chemical composition of the surface region of the sample after annealing at 1273 K changed only slightly. In the Fe 2p_3/2_ spectrum, the positions of the components were similar to the sample heated at 1073 K; however, the concentration of the metallic peak increased.

Figure 8 presents TMS spectra of the Fe_0.95_Ti_0.05_ –XPS and –OT samples. In the TMS spectrum of the as-prepared sample, four sextets related to iron atoms located in the bcc Fe–Ti phase can be seen. The spectrum measured for the sample after the XPS measurements does not change. In the case of the –OT sample heated in vacuum at 1073 K/2h, an additional doublet with IS = 0.231(22) mm/s and QS = 0.864(21) mm/s is found and can be assigned to the Fe_2_Ti compound [17]. The spectra obtained for the –XPS sample after oxidation for 1 h and 5 h reveal the presence of sextets related to Fe_2_O_3_ and Fe_3_O_4_. The determined c(Fe_2_O_3_) and c(Fe_3_O_4_) after 5 h of oxidation are equal to 5.9% and 4.7%, respectively (Table 3). In the case of the –OT sample after 1 h and 5 h of oxidation, the TMS spectra reveal the absence of sextets that correspond to iron oxides. This result indicates that high corrosion resistance is achieved for this sample, which may be comparable to those observed for the Fe–Cr and Fe–Cr–Si alloys [7,8].

### 3.5. Fe–Ge Alloy

Table 7 presents the surface concentration ratios for the Fe_0.95_Ge_0.05_ sample after annealing at the selected temperatures. Despite the fact that all estimated c_Ge_/c*_Fe_* values are higher than the bulk value, the efficiency of segregation is much lower than for other studied samples. The highest ratio was obtained for 1273 K. At the same time, the determined c*_O_*/c_*Fe+Ge*_ ratios clearly show a significant decrease in oxygen in the surface region just after the first annealing. Taking this into account, it may be stated that the surface segregation of Ge atoms in α-Fe is activated only by temperature, and this process is much less effective in comparison with Al, V, Ti, Cr and Si solutes [8].
Figure 8The room temperature TMS spectra measured for Fe_0.95_Ti_0.05_ samples: heated at 1273 K/15 min in UHV marked as –XPS sample and heated at vacuum at 1073 K/2 h marked as –OT. Samples were oxidised in air at 870 K.
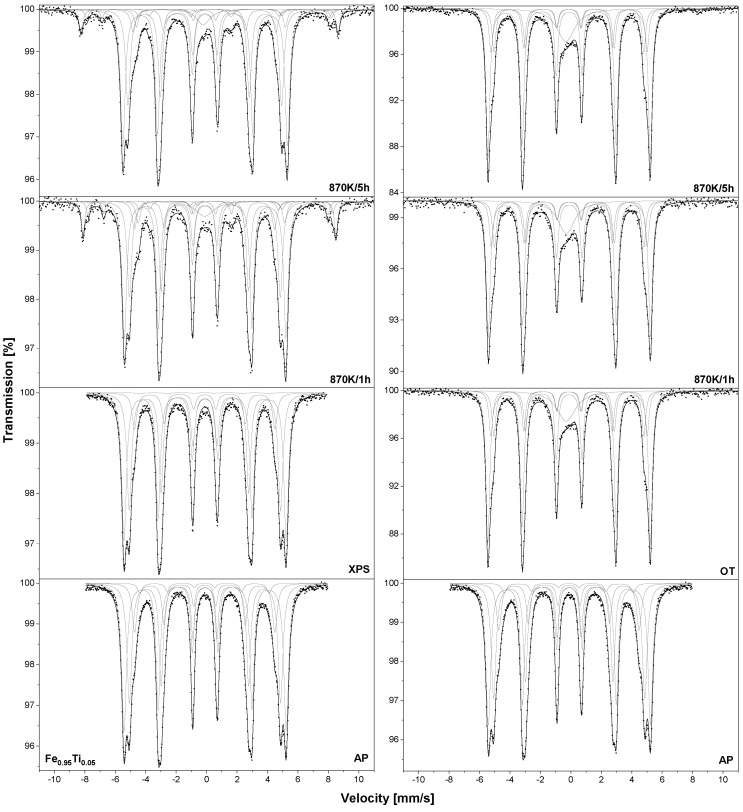


Figure 9 presents the deconvoluted XPS spectra of the Fe_0.95_Ge_0.05_ sample annealed at 1273 K. The Ge 2p and Fe 2p_3/2_ spectra are composed of pure elements and some oxides. In the case of germanium, there are Ge^0^ at the BE of 121.3 eV and oxidised Ge^4+^ located at 122.6 eV, 124.7 eV and 126.2 eV [46,47]. The iron components are located as in previous samples: the main Fe^0^ at a BE of 707.0 eV, and the Fe^2+^ and Fe^3+^ species at 708.4 eV, 710.0 eV and 711.8 eV. The C 1s spectrum reveals the presence of one peak at 285.0 eV, assigned to C–C. In the O 1s spectrum, three components can be distinguished: Fe–O at 530.8 eV, O–C at 532.3 eV and O=C at 534.0 eV.

As with the Fe_0.90_Al_0.05_V_0.05_ sample, the surface concentration of Ge increases gradually with temperature; thus, the oxidation experiment was performed only for the Fe_0.95_Ge_0.05_ sample heated at 1273 K (–XPS). The obtained TMS spectra are presented in Figure 10. For the samples before oxidation, three sextets related to iron atoms located in the bcc Fe–Ge alloy can be observed. After the oxidation process at 870 K for 5 h, additional components are observed and they can be assigned to the Fe_2_O_3_ and Fe_3_O_4_. The determined c(Fe_2_O_3_) = 8.8% and c(Fe_3_O_4_) = 6.9% indicate that the Fe_0.95_Ge_0.05_ alloy exhibits the worst anti-corrosion properties of all alloys studied in this work (see Table 3).

### 3.6. Anti-Corrosion Properties of the Studied Alloys

The XPS results show that the surface of all as-prepared samples was contaminated with oxygen and carbon. As was mentioned above, this result was expected because the samples, before being placed in the UHV chamber, had direct contact with atmospheric gases. After heat treatment, the concentration of oxygen and carbon decreases, while the surface concentration of solutes increases. In the case of the Fe_0.95_Ge_0.05_ and Fe_0.90_Al_0.05_V_0.05_ samples, the c_*solutes*_/c*_Fe_* ratios increase and c*_O_*/c_*(Fe+solutes)*_ ratios gradually decrease with increasing annealing temperature. This finding suggests that in those alloys, the surface segregation processes of the solutes are mainly governed by temperature. At the same time, for the Fe_0.95_Al_0.05_, Fe_0.95_V_0.05_ and Fe_0.95_Ti_0.05_ samples, one can observe the optimal temperature at which heat treatment leads to the highest concentration of solutes on the surface. In this case, the oxygen-induced surface segregation process can be considered. One of the most notable examples—the fact that the alloy surface composition depends not only on temperature but also on environmental parameters (i.e., oxygen atoms interact with the surface atoms)—is reported in [48]. The authors showed that for various polycrystalline alloys such as PtNi, PdAu, PdCu and CuFe, the surface segregating components, in the order given before, were Pt, Au, Cu and Cu (without oxygen influence) and Ni, Pd, Pd and Fe (with oxygen influence) [48]. More recently, Tafen et al. in [49] reported that yttrium in a ternary CuPdY alloy does not surface segregate in vacuum. However, in the presence of oxygen, Y preferentially occupies surface sites due to its stronger oxygen affinity compared to Cu and Pd. In the case of iron alloys, the oxygen-induced surface segregation process was previously observed in the Fe–Cr and Fe–Cr–Si systems [8], where the surface segregation of Cr and Si atoms was greatly enhanced by the presence of oxygen atoms adsorbed at the surface of the samples. Therefore, the XPS results obtained fo the Fe_0.95_Al_0.05_, Fe_0.95_V_0.05_ and Fe_0.95_Ti_0.05_ samples suggest that in the presence of oxygen, Al, V or Ti atoms replace Fe atoms in iron oxides located at the alloy surface. A plausible explanation is that the remaining Fe atoms are incorporated in the bcc phase, resulting in a decrease in the overall Fe concentration on the surface, as observed with the XPS analysis.
Figure 10The room temperature TMS spectra measured for Fe_0.95_Ge_0.05_ sample heated at 1273 K/15 min in UHV marked as –XPS. Sample was oxidised in air at 870 K.
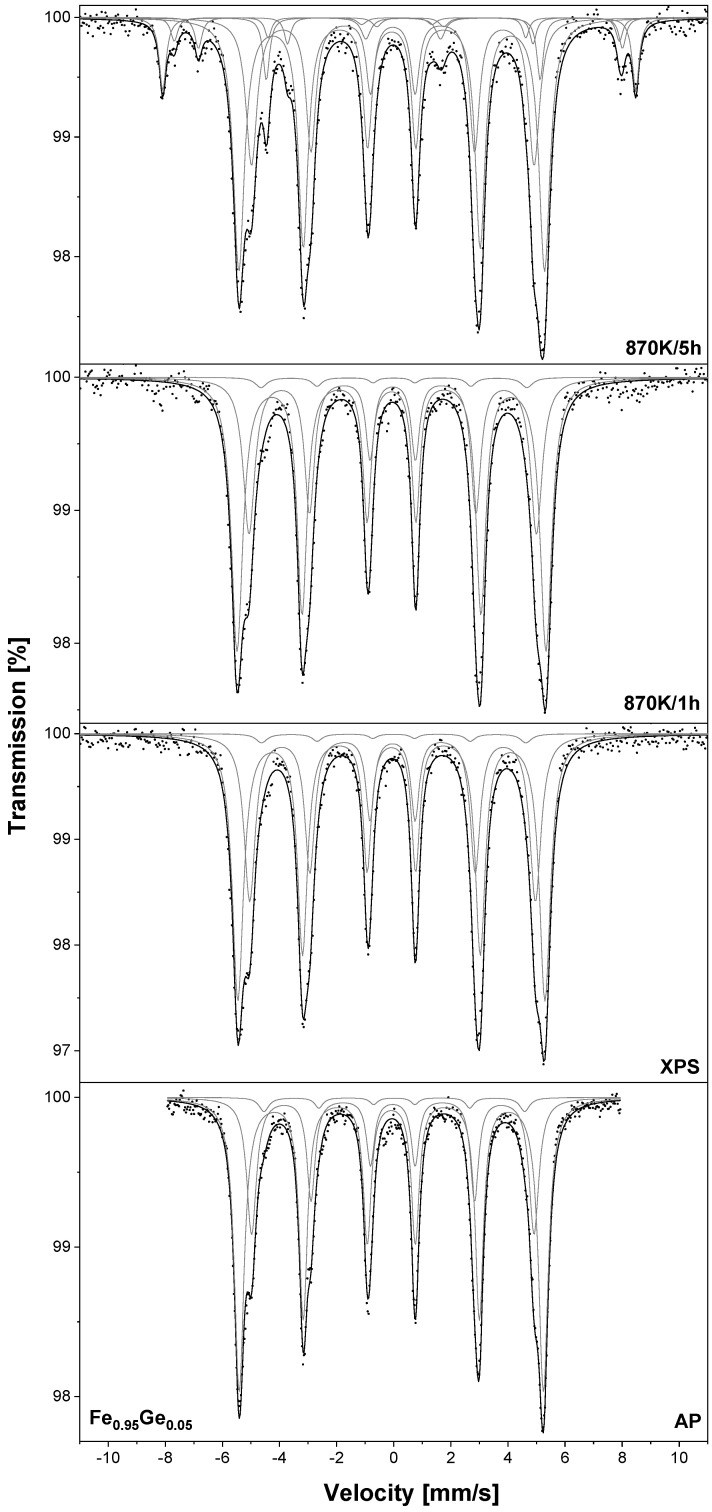



The results presented in Table 3 reveal that, among all alloys studied in this work, the worst anti-corrosion properties were exhibited by Fe_0.95_Ge_0.05_. This finding can be explained by taking into account the standard Gibbs free energy changes $\Delta G^{O}$ of various oxidation reactions at 870 K, which are presented in Table 8. Since the values of $\Delta G^{O}$ calculated for Ge and Fe oxides are comparable to each other, in the case of the Fe–Ge alloy exposed to air at elevated temperature, it is plausible to assume that iron and germanium atoms oxidise simultaneously and a passive layer will not be formed. Moreover, according to data presented in Table 7, the surface concentration of Ge atoms is relatively low and only slightly higher than the bulk value for all annealing temperatures. This may suggest that the positive influence of adsorbed oxygen atoms on the efficiency of the surface segregation process can only be expected in iron alloys, where solute atoms have a much lower Gibbs free energy of oxidation than iron atoms.

In the case of iron-based alloys with Al and V solutes, the results obtained reveal relatively strong surface segregation of the solutes in these systems. In the studied Fe_0.95_Al_0.05_, Fe_0.95_V_0.05_ and Fe_0.90_Al_0.05_V_0.05_ samples, we obtained, after heat treatment, relatively high c*_Al_*/c*_Fe_* and c*_V_*/c*_Fe_* ratios, which are between one and two. These ratios are comparable to c*_Cr_*/c*_Fe_* and c*_Si_*/c*_Fe_*, the ones obtained for the Fe–Cr and Fe–Cr–Si alloys [7,8]. Since the $\Delta G^{O}$ values of V_2_O_3_ and Al_2_O_3_ are lower than that calculated for Cr_2_O_3_, one can expect that the alloys covered by a large amount of Al and V atoms should exhibit corrosion resistance much better than Fe–Cr and Fe–Cr–Si. Surprisingly, the oxidation experiment does not confirm that. In all alloys with Al and V solutes, which were exposed to air at 870 K for 5 h, the presence of iron oxides was detected. In particular, for Fe_0.90_Al_0.05_V_0.05_ after 5 h of oxidation, c(Fe_2_O_3_) = 2.9%, while in the case of Fe_0.90_Cr_0.05_Si_0.05_, the presence of iron oxides was not detected even after exposure to air at 870 K for 88 h [8]. This finding suggests that the passive layer formed on the surface of the Fe_0.95_Al_0.05_, Fe_0.95_V_0.05_ and Fe_0.90_Al_0.05_V_0.05_ alloys is less stable and effective against corrosion at high temperatures compared to Fe–Cr and Fe–Cr–Si.

The results obtained for the Fe_0.95_Ti_0.05_ sample indicate that the addition of Ti atoms to α-Fe significantly slows the oxidation process of iron atoms during exposure of the alloy to air at high temperature. The absence of iron oxides in the –OT sample oxidised for 5 h at 870 K can be connected with the extremely high level of Ti surface concentration obtained for that sample (c*_Ti_*/c*_Fe_* = 4.34) and $\Delta G^{O}$ of TiO_2_, which is almost two times lower than the value determined for iron oxides. Unlike other alloys studied in this work, the passive layer formed on the surface of Fe_0.95_Ti_0.05_ greatly enhances its corrosion resistance. However, it should be noted that the heat treatment of the as-prepared Fe_0.95_Ti_0.05_ alloy causes the formation of an Fe_2_Ti phase, which could have a negative influence on the mechanical properties of the sample studied. These findings can be compared with results reported by Jimbo et al. in [50], where the behaviour of Ti in Fe-0.29, 0.64 and 1.48 wt%Ti alloys was investigated using a time-of-flight atom-probe. They reported three important findings: (i) Ti segregates to the surface when Fe-Ti samples are heated at and above 870 K in vacuum; (ii) chemisorption-induced surface segregation using H_2_, D_2_ and CO was not observed; (iii) when heated in O_2_ gas, two distinctly different types of oxidation were observed depending upon O_2_ partial pressure. At the lower O_2_ pressure (10^−9^ Torr), Ti oxide scale was formed on the surface, which markedly slowed further oxidation of the alloy substrate, while at a higher O_2_ pressure (10^−5^ Torr), subsurface oxidation occurred, forming Fe oxides and Ti-rich clusters within the alloy matrix. Our results confirm Ti surface segregation and additionally show that chemisorption-induced surface segregation using O_2_ occurs. In the case of oxidation of Fe-Ti alloys at high O_2_ pressure, our sample does not reveal the presence of iron oxides. However, our sample contains many more Ti atoms than the samples studied in [50].

Finally, it should be noted that in the case of the Fe_0.95_Al_0.05_, Fe_0.95_V_0.05_ and Fe_0.95_Ti_0.05_ alloys, where the oxidation experiment was performed for samples annealed at the optimal temperature for surface segregation of solutes as well as for the samples annealed at the highest temperature of 1273 K, the results obtained clearly show that the former have much better anti-corrosion properties. Therefore, it can be stated that the corrosion resistance of these alloys is directly connected with the surface concentration of the Al, V and Ti solutes.

## 4. Conclusions

The surface segregation process of solutes and its influence on the anti-corrosion proprieties of Fe_0.95_Al_0.05_, Fe_0.95_V_0.05_, Fe_0.90_Al_0.05_V_0.05_, Fe_0.95_Ti_0.05_ and Fe_0.95_Ge_0.05_ were investigated. It was found that the Fe_0.95_Ti_0.05_ sample annealed at 1073 K under vacuum has much better anti-corrosion properties than the other alloys studied. Despite the fact that this sample contains only 5 at.% of Ti, the XPS measurements reveal that the surface concentration of titanium is more than four times higher than that of iron. This effect can be connected with the strong surface segregation of Ti and is mainly responsible for the good corrosion resistance of the material. In the case of other alloys, the obtained surface concentrations of solutes are much lower. Therefore, one can conclude that among the Al, V, Ti and Ge solutes considered, only Ti atoms significantly improve the corrosion resistance of Fe-based alloys, and this effect may be comparable with that observed previously for Cr and Si solutes. The reported results pave the way for further research investigating the anti-corrosion properties of iron-based ternary Fe–Cr–Ti and Fe–Si–Ti alloys or even quaternary Fe–Cr–Si–Ti alloys.

## Figures and Tables

**Figure 1 materials-18-00557-f001:**
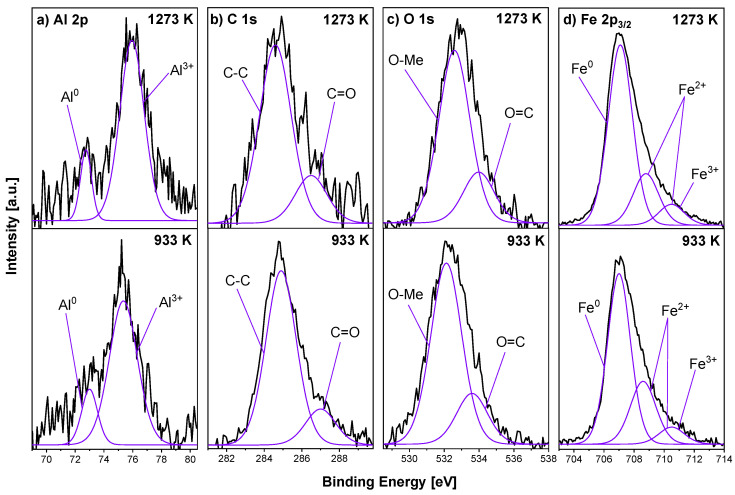
Selected XPS spectra—(**a**) Al 2p, (**b**) C 1s, (**c**) O 1s and (**d**) Fe 2p_3/2_—for Fe_0.95_Al_0.05_ samples annealed in UHV at 933 K and 1273 K/15 min.

**Figure 2 materials-18-00557-f002:**
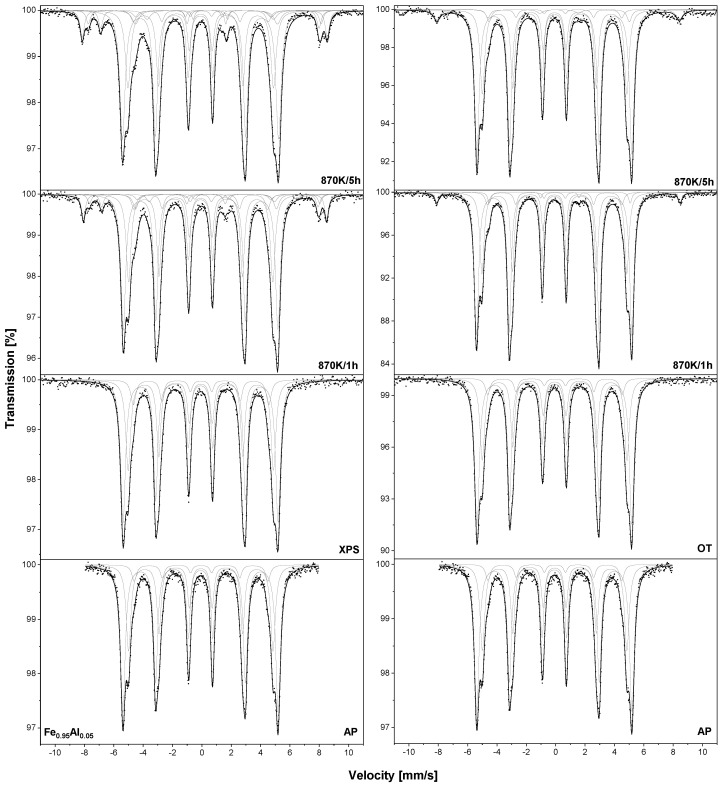
The room temperature TMS spectra measured for Fe_0.95_Al_0.05_ samples—those heated at 1273 K/15 min in UHV marked as –XPS sample and those heated in vacuum at 933 K/2 h marked as –OT. Samples were oxidised in air at 870 K.

**Figure 3 materials-18-00557-f003:**
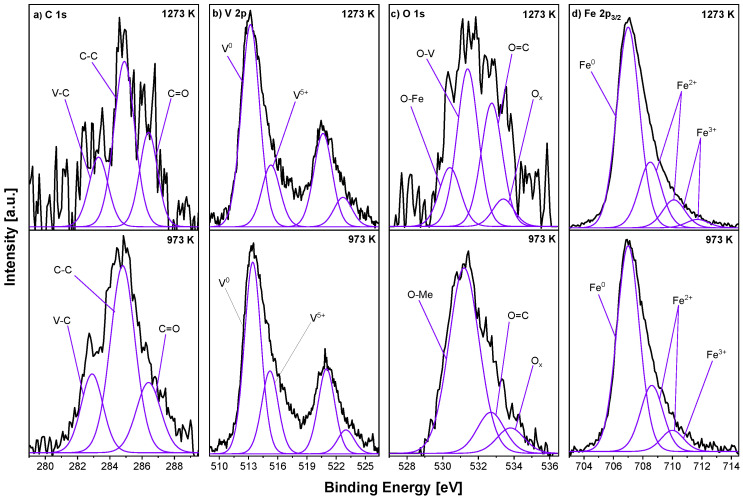
Selected XPS spectra—(**a**) C 1s, (**b**) V 2p, (**c**) O 1s and (**d**) Fe 2p_3/2_—for Fe_0.95_V_0.05_ samples and those annealed in UHV at 973 K and 1273 K/15 min.

**Figure 4 materials-18-00557-f004:**
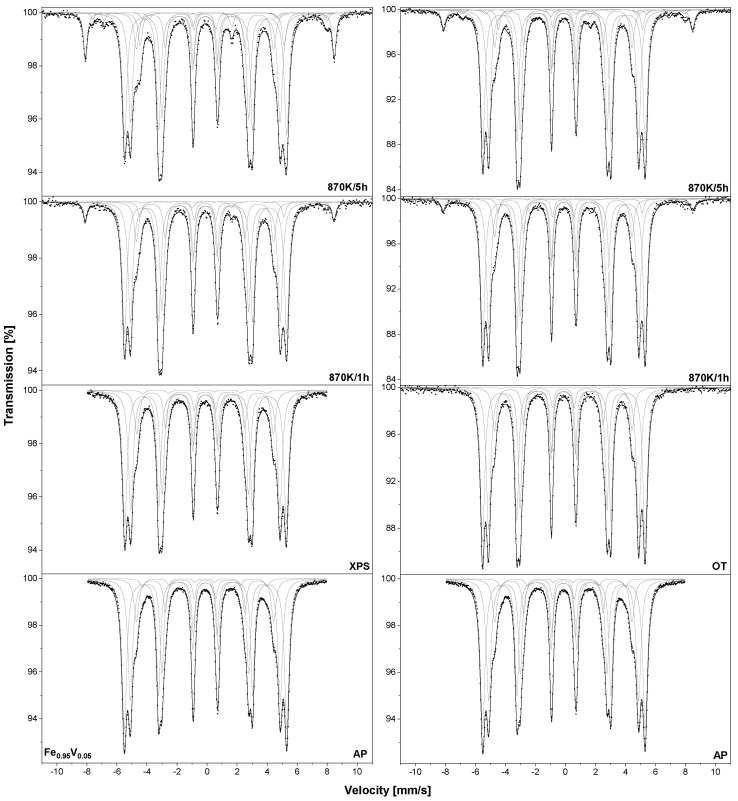
The room temperature TMS spectra measured for Fe_0.95_V_0.05_ samples: heated at 1273 K/15 min in UHV marked as –XPS sample and heated at vacuum at 973 K/2 h marked as –OT. Samples were oxidised in air at 870 K.

**Figure 5 materials-18-00557-f005:**
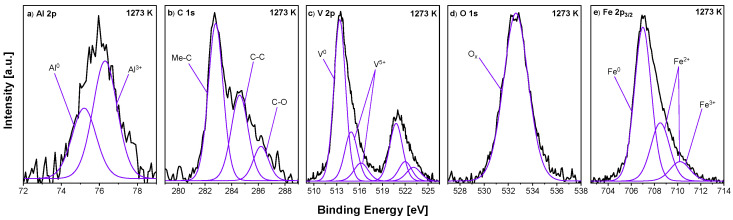
Selected XPS spectra—(**a**) Al 2p, (**b**) C 1s, (**c**) V 2p, (**d**) O 1s and (**e**) Fe 2p_3/2_—for Fe_0.90_Al_0.05_V_0.05_ sample annealed in UHV at 1273 K/15 min.

**Figure 6 materials-18-00557-f006:**
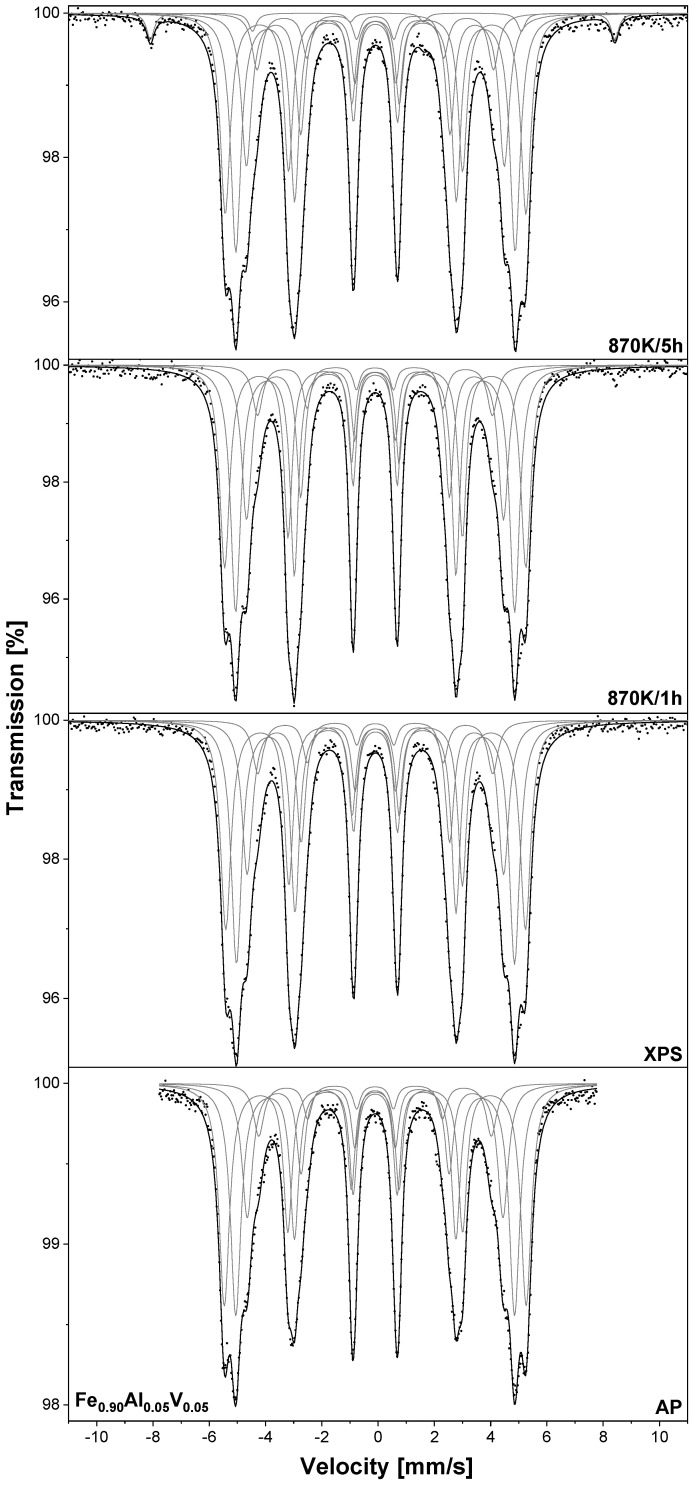
The room temperature TMS spectra measured for Fe_0.90_Al_0.05_V_0.05_ sample heated at 1273 K/15 min in UHV marked as –XPS. Sample was oxidised in air at 870 K.

**Figure 7 materials-18-00557-f007:**
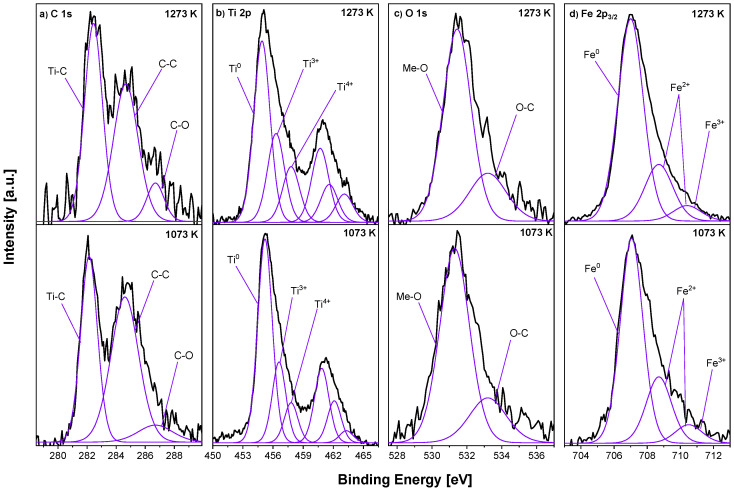
Selected XPS spectra—(**a**) C 1s, (**b**) Ti 2p, (**c**) O 1s and (**d**) Fe 2p_3/2_—for Fe_0.95_Ti_0.05_ samples annealed in UHV at 1073 K and 1273 K/15 min.

**Figure 9 materials-18-00557-f009:**
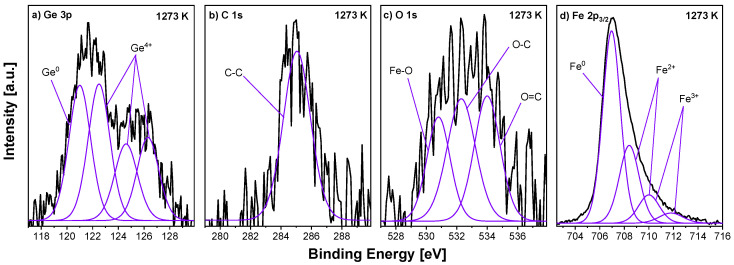
Selected XPS spectra—(**a**) Ge 3p, (**b**) C 1s, (**c**) O 1s and (**d**) Fe 2p_3/2_—for Fe_0.95_Ge_0.05_ samples annealed in UHV at 1273 K/15 min.

**Table 1 materials-18-00557-t001:** Hyperfine parameters of ^57^Fe nuclei in the investigated alloys.

Sample	*B* _0_	Δ *B* _1_	Δ *B* _2_	*IS* _0_	Δ *IS* _1_	Δ *IS* _2_
	**[T]**	**[T]**	**[T]**	**[mm/s]**	**[mm/s]**	**[mm/s]**
Fe–Al	32.8(1)	−2.3(2)	0	0.017(1)	−0.019(2)	0
Fe–V	33.5(1)	−2.6(1)	−2.6(1)	0.009(1)	−0.017(1)	−0.017(1)
Fe–Al–V	33.0(3)	−2.4(3)	−2.4(3)	0.021(2)	−0.006(2)	−0.006(2)
Fe–Ti	33.5(2)	−2.2(3)	−2.2(3)	0.014(2)	−0.021(3)	−0.021(3)
Fe–Ge	33.2(3)	−2.5(5)	0	0.037(3)	0.038(5)	0

**Table 2 materials-18-00557-t002:** Surface concentration ratios determined from XPS measurements for the Fe_0.95_Al_0.05_ alloy. At each step, the sample was heated in the UHV for 15 min. SBR denotes the signal-to-background ratio of the selected XPS spectrum.

Heating Temperature	$c_{\mathrm{Al}}/c_{\mathrm{Fe}}$	$c_{O}/c_{\left(\mathrm{Fe}+\mathrm{Al}\right)}$
as-prepared	low SBR	12.69
823 K	low SBR	4.26
873 K	1.01	0.47
933 K	1.13	0.43
973 K	1.07	0.33
1073 K	0.86	0.24
1173 K	0.70	0.21
1273 K	0.57	0.21
bulk	0.05	–

**Table 3 materials-18-00557-t003:** Fractions of absorbing ^57^Fe atoms located in iron oxides in investigated alloys. Measurement uncertainties are not greater than 1%.

Sample		Time [h]	c(Fe_2_O_3_) [%]	c(Fe_3_O_4_) [%]
Fe–Al	–XPS	1 h	6.5	6.1
		5 h	7.4	8.8
	–OT	1 h	3.4	0
		5 h	5.2	0
Fe–V	–XPS	1 h	4.5	0
		5 h	10.1	3.5
	–OT	1 h	3.7	0
		5 h	5.3	1.3
Fe–Al–V	–XPS	1 h	0	0
		5 h	2.9	0
Fe–Ti	–XPS	1 h	7.3	6.0
		5 h	5.9	4.7
	–OT	5 h	0	0
Fe–Ge	–XPS	5 h	8.8	6.9

**Table 4 materials-18-00557-t004:** Surface concentration ratios determined from XPS measurements for Fe_0.95_V_0.05_ alloy. At each step, sample was heated in UHV for 15 min.

Heating Temperature	$c_{V}/c_{\mathrm{Fe}}$	$c_{O}/c_{\left(\mathrm{Fe}+V\right)}$
as-prepared	0.48	5.39
823 K	0.42	1.16
873 K	0.47	0.51
933 K	1.10	0.30
973 K	1.25	0.30
1073 K	1.15	0.26
1173 K	0.65	0.19
1273 K	0.38	0.05
bulk	0.05	–

**Table 5 materials-18-00557-t005:** Surface concentration ratios determined from XPS measurements for the Fe_0.90_V_0.05_Al_0.05_ alloy. At each step, the sample was heated in the UHV for 15 min. SBR denotes the signal-to-background ratio of the selected XPS spectrum.

Heating Temperature	$c_{\mathrm{Al}}/c_{\mathrm{Fe}}$	$c_{V}/c_{\mathrm{Fe}}$	$c_{O}/c_{\left(\mathrm{Fe}+\mathrm{Al}+V\right)}$
as-prepared	low SBR	0.31	7.37
823 K	0.31	0.33	1.25
873 K	1.24	0.48	0.68
933 K	1.41	0.63	0.65
973 K	1.49	0.79	0.54
1073 K	1.20	0.72	0.43
1173 K	1.14	0.85	0.38
1273 K	1.46	1.87	0.35
bulk	0.06	0.06	–

**Table 6 materials-18-00557-t006:** Surface concentration ratios determined from XPS measurements for the Fe_0.95_Ti_0.05_ alloy. At each step, the sample was heated in the UHV for 15 min. SBR denotes the signal-to-background ratio of selected XPS spectrum.

Heating Temperature	$c_{\mathrm{Ti}}/c_{\mathrm{Fe}}$	$c_{O}/c_{\left(\mathrm{Fe}+\mathrm{Ti}\right)}$
as-prepared	low SBR	22.79
823 K	0.07	3.83
873 K	0.57	1.03
933 K	2.42	0.85
973 K	3.90	0.68
1073 K	4.34	0.52
1173 K	1.40	0.37
1273 K	0.68	0.28
bulk	0.05	–

**Table 7 materials-18-00557-t007:** Surface concentration ratios determined from XPS measurements for Fe_0.95_Ge_0.05_ alloy. At each step, sample was heated in UHV for 15 min. SBR denotes signal–to–background ratio of selected XPS spectrum.

Heating Temperature	$c_{\mathrm{Ge}}/c_{\mathrm{Fe}}$	$c_{O}/c_{\left(\mathrm{Fe}+\mathrm{Ge}\right)}$
as-prepared	low SBR	9.92
823 K	0.14	0.63
873 K	0.10	0.29
933 K	0.09	0.25
973 K	0.10	0.19
1073 K	0.11	0.12
1173 K	0.13	0.05
1273 K	0.17	0.06
bulk	0.05	–

**Table 8 materials-18-00557-t008:** The standard Gibbs free energy changes $\Delta G^{O}$ of various oxidation reactions at 870 K calculated from the Ellingham diagram [9].

Oxide	$\Delta G^{O}$ [kJ/mol $O_{2}$]
Fe_2_O_3_	−397
GeO_2_	−411
Fe_3_O_4_	−417
Cr_2_O_3_	−601
V_2_O_3_	−680
SiO_2_	−753
TiO_2_	−786
Al_2_O_3_	−934

## Data Availability

The dataset is available on request from the authors due to the internal policy.

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
