# Peer review of "Surface Segregation Process and Its Influence on High-Temperature Corrosion of Iron-Based Alloys Containing Aluminium, Vanadium, Titanium and Germanium"

_materials, 2025, doi:10.3390/ma18030557_

Round 1
Reviewer 1 Report
Comments and Suggestions for Authors
Review Report
Article title: Surface segregation process and its influence on high–temperature corrosion of iron–based alloys containing aluminium, vanadium, titanium and germanium
In this paper the authors studied the surface segregation of solutes (Al, V, Ti, and Ge) and their influence on the high-temperature corrosion of binary and ternary iron-based alloys. The alloys were annealed at 823 to 1273 K for 15 min in ultrahigh vacuum to induce the surface segregation. After annealing, the samples were oxidized at 870 K in air. The examination of surface segregation phenomena using XPS is interesting; however, the materials are insufficiently characterized. The microstructure and phase constitution are not revealed. Furthermore, the oxidation behavior is studied by spectral analysis alone. The oxidation mechanism is not discussed. The following comments must be considered:
1.Materials is a Materials Science Journal. As such, it is necessary to provide a sufficient characterization of the materials studied. You should show the microstructure of the as-prepared alloys (after arc-melting and cold-rolling). Have you observed elongated grains after rolling? What was the grain size?
2.Were the alloys metallographically prepared by grinding and polishing prior to annealing?
3.Have you recorded any weight gain during oxidation?
4.How thick was the oxide scale produced during oxidation at 870 K for 5 h?
5.Have you studied the phase constitution of the oxide scale using XRD?
6.Have you observed any Al2O3 and TiO2 scales located under the Fe2O3 scale?
7.The discussion of the oxidation mechanism should be based on Gibbs energies of formation of various oxides, i.e., the so-called Ellingham diagram. The comparison of standard electrode potentials is meaningful only when aqueous corrosion is considered. It is not the present case.
8.It is necessary to examine the cross-section of the oxide scale and discuss the oxidation mechanism.
9.It is also recommended to record the specific weight gain of the alloys as a function of time and obtain the rate constant of oxidation. Data should be compared with previously studied iron-based alloys.
Author Response
We thank the Referee for carefully reading and numerous comments. In all cases they were taken into account and discussed.
Issue 1
Materials is a Materials Science Journal. As such, it is necessary to provide a sufficient characterization of the materials studied. You should show the microstructure of the as-prepared alloys (after arc-melting and cold-rolling). Have you observed elongated grains after rolling? What was the grain size?
Discussion: In this work we have studied standard polycrystalline iron-based alloys prepared by arc-melting and cold-rolling procedures. We focused on the surface segregation and corrosion in these materials so we do not perform additional studies focused on characterization of these well-known materials.
Issue 2
Were the alloys metallographically prepared by grinding and polishing prior to annealing?
Discussion: No, the alloy samples were studied without additional processing such as grinding and polishing.
Issue 3
Have you recorded any weight gain during oxidation?
Discussion: We tried to measure the weight gain during oxidation. However, since the mass of the samples was small, we did not receive any reliable results. We believe that the measurement of Mossbauer spectra of oxidized samples gave much more reliable corrosion results.
Issue 4
How thick was the oxide scale produced during oxidation at 870 K for 5 h?
Discussion: According to TMS results, which gives fractions of absorbing 57Fe atoms located in iron oxides in the investigated alloys, the rough estimation of the oxide scale thickness produced during oxidation at 870 K for 5 h is between 0 μm (for Fe-Ti) and about 4 μm (for Fe-Ge).
Issue 5
5.Have you studied the phase constitution of the oxide scale using XRD?
Discussion: In our opinion, XRD measurements of the oxide scale is very problematic for several reasons. (1) The thickness of the oxide scale is to small for standard XRD measurement. (2) Diffraction profiles for Fe compounds exhibit high backgrounds due to fluorescent X-rays generated by Fe, which makes it difficult to distinguish low-intensity diffraction peaks from the background, resulting in errors in practically any quantitative analysis.
Issue 6
6.Have you observed any Al2O3 and TiO2 scales located under the Fe2O3 scale?
Discussion: Yes we observe the presence of Al2O3 and TiO2 oxides in our samples.
Issue 7
The discussion of the oxidation mechanism should be based on Gibbs energies of formation of various oxides, i.e., the so-called Ellingham diagram. The comparison of standard electrode potentials is meaningful only when aqueous corrosion is considered. It is not the present case.
Discussion: We agree with the Reviewer. In revised manuscript the discussion of the oxidation mechanism is based on Gibbs energies of formation of various oxides (Ellingham diagrams).
Issue 8
It is necessary to examine the cross-section of the oxide scale and discuss the oxidation mechanism.
Discussion: We believe that XPS results reveal the chemical composition of the oxide scale. In this work we just wanted to show which element added to iron would have the most beneficial effect on its corrosion properties and whether this effect is related to the surface segregation process.
Issue 9
It is also recommended to record the specific weight gain of the alloys as a function of time and obtain the rate constant of oxidation. Data should be compared with previously studied iron-based alloys.
Discussion: Again, in this work we just wanted to show which element added to iron would have the most beneficial effect on its corrosion properties and whether this effect is related to the surface segregation process.
Reviewer 2 Report
Comments and Suggestions for Authors
The paper deals with a study on the surface segregation process and its influence on high–temperature corrosion of iron–based alloys containing Al, V, Ti and Ge. The authors found that the Fe0.95Ti0.05 sample annealed at 1073K has much better anti–corrosion properties than other alloys. The author analyzes the reasons. The research has some significance. However, some problems need to be modified before final acceptance.
1. The author himself has conducted extensive research in this field and fully summarized it, but whether other people have similar studies, the summary is slightly insufficient.
2. The suthors stated that, the passive layer formed on the surface of Fe0.95Ti0.05 greatly enhances its resistance to corrosion. But what evidence can be provided in the paper for this description?
3. The author selected five alloys for research, and simply listed the XPS and TMS test results of several alloys, which seems to lack in-depth research on the corrosion structure and properties of materials.
Author Response
We thank the Referee for carefully reading and numerous comments. In all cases they were taken into account and discussed.
Issue 1
The author himself has conducted extensive research in this field and fully summarized it, but whether other people have similar studies, the summary is slightly insufficient.
Discussion: In Introduction and Results and Discussion sections we added new information regarding studies of surface segregation and selective oxidation phenomena on various alloys.
Issue 2
The Authors stated that, the passive layer formed on the surface of Fe0.95Ti0.05 greatly enhances its resistance to corrosion. But what evidence can be provided in the paper for this description?
Discussion: Our suggestion is based on some indirect evidences. As it was presented in the manuscript, the surface concentration of Ti atoms in Fe0.95Ti0.05 alloy which was annealed at optimal temperature was much higher than for the same alloy but annealed at 1273 K. At the same time, the first one exhibits much better corrosion properties. Therefore, it is plausible to assume that high concentration of Ti at the alloys surface greatly enhance its corrosion resistance. Similar results were obtained for Fe-V and Fe-Al alloys.
Issue 3
The author selected five alloys for research, and simply listed the XPS and TMS test results of several alloys, which seems to lack in-depth research on the corrosion structure and properties of materials.
Discussion: In this work we just wanted to show which element added to iron would have the most beneficial effect on its corrosion properties and whether this effect is related to the surface segregation process. In revised manuscript the discussion of the oxidation mechanism is based on Gibbs energies of formation of various oxides (Ellingham diagrams).
Round 2
Reviewer 1 Report
Comments and Suggestions for Authors
Review Report
Article title: Surface segregation process and its influence on high–temperature corrosion of iron–based alloys containing aluminium, vanadium, titanium and germanium
The authors have not answered my previous comments adequately. Therefore, I cannot recommend their manuscript for publication. The reasons for rejecting the paper are given below:
1.There is no microstructure of the starting materials presented in the paper. You claim that you had “standard polycrystalline iron-based alloys prepared by arc-melting and cold-rolling procedures”. What was the grain size? How did their microstructure change after cold rolling? You must support your claims by presenting SEM images.
2.The authors refer the readers to their previous articles (Ref. [8], [25]). However, no microstructure of the materials is presented there either.
3.The authors say that “the alloy samples were studied without additional processing such as grinding and polishing”. How is it possible? Iron-based alloys have a naturally occurring oxide layer on the surface. It must be removed by grinding and polishing prior to any oxidation experiment. Furthermore, it is necessary to eliminate the surface roughness. Corrosion is a surface reaction. Therefore, you must eliminate the surface roughness and define the surface area properly.
4.The authors claim that according to Mössbauer spectra “the rough estimation of the oxide scale thickness produced during oxidation at 870 K for 5 h is between 0 μm (for Fe-Ti) and about 4 μm (for Fe-Ge)”. These claims must be supported by the presentation of the SEM images of cross sections of the specimens.
5.You claim that you observed the presence of Al2O3 and TiO2 oxides in your samples. The XRD records should be presented in the paper to confirm these phases. If the scale is thin, it is possible to run a grazing-incident XRD experiment. It will increase the area of the oxide layer through which the X-ray beam passes through. Furthermore, the XRD patterns of the starting materials should be shown in the paper as well for the sake of comparison.
6.In conclusion, the authors studied the oxidation behavior by employing spectral analyses alone. The oxidation behavior, however, is usually studied by a combination of techniques including microscopy imaging, X-ray diffraction and thermogravimetry. Additionally, the samples have not been properly metallographically prepared by grinding and polishing prior to experiments. Therefore, the XPS results might have been significantly affected by the presence of naturally occurring oxides. I cannot recommend this paper for publication. The authors must adapt standard procedures to study the oxidation behavior of the alloys. Furthermore, it is necessary to inspect the materials using SEM/EDS and XRD to reveal the microstructure and phase constitution of the oxide layer. Otherwise, the paper cannot be published in a standard Materials Science journal.
Author Response
We thank the Referee for carefully reading and numerous comments.
All issues
There is no microstructure of the starting materials presented in the paper. You claim that you had “standard polycrystalline iron-based alloys prepared by arc-melting and cold-rolling procedures”. What was the grain size? How did their microstructure change after cold rolling? You must support your claims by presenting SEM images.
The authors refer the readers to their previous articles (Ref. [8], [25]). However, no microstructure of the materials is presented there either.
The authors say that “the alloy samples were studied without additional processing such as grinding and polishing”. How is it possible? Iron-based alloys have a naturally occurring oxide layer on the surface. It must be removed by grinding and polishing prior to any oxidation experiment. Furthermore, it is necessary to eliminate the surface roughness. Corrosion is a surface reaction. Therefore, you must eliminate the surface roughness and define the surface area properly.
The authors claim that according to Mössbauer spectra “the rough estimation of the oxide scale thickness produced during oxidation at 870 K for 5 h is between 0 μm (for Fe-Ti) and about 4 μm (for Fe-Ge)”. These claims must be supported by the presentation of the SEM images of cross sections of the specimens.
You claim that you observed the presence of Al2O3 and TiO2 oxides in your samples. The XRD records should be presented in the paper to confirm these phases. If the scale is thin, it is possible to run a grazing-incident XRD experiment. It will increase the area of the oxide layer through which the X-ray beam passes through. Furthermore, the XRD patterns of the starting materials should be shown in the paper as well for the sake of comparison.
In conclusion, the authors studied the oxidation behavior by employing spectral analyses alone. The oxidation behavior, however, is usually studied by a combination of techniques including microscopy imaging, X-ray diffraction and thermogravimetry. Additionally, the samples have not been properly metallographically prepared by grinding and polishing prior to experiments. Therefore, the XPS results might have been significantly affected by the presence of naturally occurring oxides. I cannot recommend this paper for publication. The authors must adapt standard procedures to study the oxidation behavior of the alloys. Furthermore, it is necessary to inspect the materials using SEM/EDS and XRD to reveal the microstructure and phase constitution of the oxide layer. Otherwise, the paper cannot be published in a standard Materials Science journal.
Discussion:
In our opinion the Reviewer slightly misunderstands the main topic of this work. In our work we focused on the observation of the surface segregation phenomenon in studied alloys. Indeed, we also additionally present the results of oxidation of these materials at high temperature in atmospheric gases. However, this part of the study is based on a comparison between the samples before and after obtaining the optimal surface concentration of solutes due to surface segregation. Since we only compare the oxidation results between the same samples but with different surface chemical composition, in our opinion the microstructure characterization is not necessary. Both samples with the optimal surface segregation and without it have the same grain sizes and shapes. TMS spectra clearly shows that all as-cast samples are bcc solid solutions. No other phases were detected. Since the samples of each alloy differ only in the surface chemical composition we can show that higher concentration of solutes at the alloys surface has a positive effect on their anti-corrosion properties or not.
Due to the fact that we also study the oxygen induced surface segregation phenomenon, extra polishing is excluded since we need the adsorbed oxygen at the alloys surface.
We do not understand why the Reviewer insists that our estimation of the oxide scale thickness must be supported by the presentation of the SEM images of cross sections of the specimens. From TMS spectra we can clearly calculate the fraction of iron oxides in oxidized samples. Since the oxidation process begins at the surface and then the oxygen atoms diffuse into deeper layers of the material we can estimate the oxide scale thickness from measured TMS spectra.
The same is with the presence of Al2O3 and TiO2 oxides in your samples. We confirm the presence of these oxides using XPS spectra. In our opinion XPS is more sensitive than XRD, especially in the case of thin layers.
Reviewer 2 Report
Comments and Suggestions for Authors
The revised paper can be accepted.
Author Response
We thank the Referee for carefully reading and positive decision.